# Microwave Simulation Experiments on Regolith (Lunar Dust) Deposition on Stainless Steel

**DOI:** 10.3390/ma14216472

**Published:** 2021-10-28

**Authors:** Nina N. Skvortsova, Vladimir D. Stepakhin, Andrey A. Sorokin, Dmitry V. Malakhov, Namik G. Gusein-zade, Nailya S. Akhmadullina, Valentin D. Borzosekov, Elena V. Voronova, Oleg N. Shishilov

**Affiliations:** 1Prokhorov General Physics Institute of the Russian Academy of Sciences, 119991 Moscow, Russia; stepakhin@fpl.gpi.ru (V.D.S.); 89199945791@mail.ru (D.V.M.); ngus@mail.ru (N.G.G.-z.); tinborz@gmail.com (V.D.B.); woronowa.elena@gmail.com (E.V.V.); 2Institute for Laser and Plasma Technologies, National Research Nuclear University MEPhI, 115409 Moscow, Russia; 3Institute of Applied Physics, Russian Academy of Sciences, 603950 Nizhny Novgorod, Russia; asorok@appl.sci-nnov.ru; 4Baikov Institute of Metallurgy and Material Science, Russian Academy of Sciences, 119334 Moscow, Russia; nakhmadullina@mail.ru; 5Faculty of Physics and Mathematics and Natural Sciences, Peoples Friendship University of Russia (RUDN University), 117198 Moscow, Russia; 6Institute of Fine Chemical Technologies, MIREA—Russian Technological University, 119454 Moscow, Russia; oshishilov@gmail.com

**Keywords:** simulation experiments, gyrotron discharge, modification of stainless steel samples, regolith

## Abstract

In this article, results are presented of experiments on depositing charged particles, which imitate the levitating dust on the Moon, on stainless steel. Ensembles of particles are created above the surface of laboratory regolith whose composition and particle size distribution imitate the dust that covers the Moon’s surface. Under the action of the gyrotron radiation on regolith, non-linear physical-chemical processes develop (breakdown, chain plasmachemical reactions, and particle scattering by the Coulomb mechanism), which lead to the appearance of a levitating cloud of particles. The simulation experiment is based on the similarity between the processes that develop in the laboratory experiments with regolith and the processes that occur on the Moon during its bombardment by micrometeorites. The effect of the levitating cloud on stainless steel plates is studied and it is shown that regolith particles in the shape of spheroids of different sizes are deposited on the surface of the plates. The dimensions of the deposited particles and the density of their placement depend on the quality of treatment of the plate surface. It is shown that the laboratory-produced dusty plasma can be used in simulation experiments to study the modification of surfaces of different materials for space technology.

## 1. Introduction

After the discovery and identification [1,2] of plasma-dust clouds above the Moon’s surface, the interest in studying these objects continuously grows. The academic interest consists in the possibility of new observations and studies of dusty plasma above the Moon’s surface during future Moon landing missions. The practical interest is determined by the importance of understanding the physical processes of interaction of the lunar dust and dusty plasma with various construction materials for the design of the equipment of Moon landing modules and also the effects of the dust on the human body [3,4,5,6,7]. Charged microparticles on the Moon have increased adhesion ability, which introduces various limitations to the use of space systems on its surface, from the contamination of the surface of solar panels to the decreased operational life of the details of rubbing mechanisms [8,9]. This is why it is currently important to obtain flows and levitating clouds of charged dust similar to lunar dust under laboratory conditions [10,11,12,13,14] to conduct simulation experiments and testing of both materials and components of future space technology, including lunar habitats [15,16]. Such experiments are conducted using acceleration technology [17,18,19,20], electron beams [12,13,14] or electrostatic powder dispensers [21]. These and other techniques are applied to imitate micrometeorites and man-made particles in simulation experiments with spacecraft materials [22,23,24] and in dust mitigation tests [25,26,27]. However, it is difficult to use such injectors to create clouds of microparticles that imitate not only the chemical composition but also size distribution of lunar dust and use such clouds to process large areas of material surface. 

In the contemporary view, the plasma-dust clouds above the Moon’s surface consist of charged particles of lunar regolith. Thousands of regolith samples were transported to the Earth, and their physical and chemical properties are described in detail in catalogues [28,29]. The density of regolith that is composed of different oxides (aluminum, silicon, iron, magnesium, etc.) is in the range 1.3–3.1 g/cm^3^. Samples of regolith also contain metal particles, e.g., some of them contain up to 1% of iron [30]. The average size of regolith particles is 70–200 µm (the granulometric composition of lunar regolith averaged over many samples can be found in [31]). It is important to note that the particle size distribution of regolith differs from the random Gaussian distribution in that it contains excess numbers of particles with larger sizes [28].

By contrast with the well-described and well-studied regolith dust collected on Moon’s surface, there are much less data on the parameters of lunar dusty plasma. To date, its chemical composition; shape, charge, and size of charged grains and their distribution over altitudes above the Moon’s surface were not described. Levitating charged dust has always been observed from satellites and during Moon landing missions, therefore, the height to which dust rises during the landing of the Moon module and an astronaut walking on the Moon’s surface in a space suit and riding a lunar rover are described.

At the same time, some specific features of the behavior of lunar dust during its interaction with ionizing radiation, charged surfaces, and photoemission electrons were only determined by simulation in laboratory experiments. In these experiments, it was shown that dust levitates above the surface that is under constant potential [32] and the charge of microparticles from different materials, i.e., imitators of lunar regolith, was studied during contact with a negatively charged metal surface [33]. For example, it was shown that 125–150 µm particles of lunar regolith were charged to 10^6^ elementary charges during contact with a metal surface with potential of –20 V [34]. The properties of lunar dust clouds, namely, the distribution of particle ensembles by altitude, size, charge, and velocity, to date, have been estimated, mainly, by physical–mathematical models. The model of the formation of photoinduced plasma-dust layer near the Moon’s surface [35] that accounts for the electron and ion flows of the solar wind and the distribution of photoelectrons in the surface layer allow one to describe the lifting of small regolith particles (about 1 µm) to the height of several meters. In articles [36,37,38], a model was presented of the appearance of larger-sized particles in the plasma-dust system of the Moon that additionally accounts for the shock waves that are caused by the fall of meteorites. The Moon is constantly bombarded by micrometeorites. During missions, the frequency of their landing was measured in the LEAM experiment. Meteorites hitting the Moon’s surface are also observed from the Earth through telescopes in the form of bright bursts [38]. Around the epicenter of the micrometeorite’s impact, in the surface layer of lunar regolith, a cascade of processes takes place besides the generation of shock waves: evaporation, melting, destruction of particles, and irreversible and elastic deformation of particles. Currently, not all of these processes are accounted for in models.

We have proposed an additional mechanism of the processes that can develop during the impact of micrometeorites with the Moon’s surface in the region of micrometeorite destruction. Two processes, the chain plasmachemical reactions above the regolith surface and scattering of charged particles from the surface by the Coulomb mechanism, can lead to additional charging of dust grains, their lifting and levitation (even from Earth, the fall of such meteoroids can clearly be seen. https://www.youtube.com/watch?v=EDDT84JlFrg (accessed on 1 September 2021)) [39]. These physical-chemical processes were experimentally discovered during the action of pulsed microwave radiation of a powerful gyrotron on different mixtures of metals and dielectrics (and in particular, regolith) and led to the appearance and levitation of ensembles of charged particles above the surface of powders [40,41]. (Note that, in lunar regolith, besides the oxide powders (dielectrics), small quantities of metal powders are also present [30], moreover, metals appear in the free state when the oxides are destroyed during the impact of meteorites with the lunar surface.) The microwave energy absorbed by the powder mixture that is required for the development of such processes is lower than the energy released during the impact of even small meteorites (micrometerorites) with the Moon’s surface.

In [42], it was shown that a cloud of microparticles is generated above the surface of powders whose chemical composition and size distribution of particles are similar to lunar regolith (with an addition of a small amount of metal powder). Experiments were carried out during the action on laboratory regolith (henceforth, we call such powder mixture created in the laboratory simply “regolith”) of gyrotron pulses with power up to 400 kW and duration up to 10 ms. In these experiments, clouds of particles levitated in the plasmachemical reactor at height up to 50 cm above the surface of powders during several seconds after gyrotron switch-off. The chemical composition, average size and size distribution of particles deposited on the reactor walls after levitation repeated those of the initial regolith mixture; however, their shape changed: where the initial particles had sharp edges, the secondary particles had a melted spheroidal shape. We proposed a method for creating levitating clouds of charged particles for simulation experiments on the action of lunar dust on different surfaces (for space and lunar tests of materials under earthbound conditions) [43]. Note that, while our use of microwave radiation to simulate micrometeorites bombardment of the regolith layer is unique, microwaves are also used in experiments on sintering of lunar regolith and its simulants [44,45] and in studies of their dielectric properties [46,47]. 

In this article, it was shown that microwave simulation experiments on regolith deposition on stainless steel can be conducted under Earth’s conditions and the effect of levitating particles of lunar regolith, generated by powerful gyrotron radiation, on the surface of stainless steel plates was studied for the first time.

## 2. Materials and Methods

### 2.1. Materials

In experiments, 10 × 400 mm rectangular supporting plates with thickness of 0.4 mm made from 309 L (ISO 3581) stainless steel were used [48]. One half of the samples were preliminarily treated by a low-temperature plasma jet of dielectric barrier discharge (DBR) [49,50]. A layer of regolith powder with thickness 0.5–0.7 mm was poured on the supporting plate of the reactor, and its upper surface was left free (the layer was not beaten down). The laboratory regolith consisted of a mixture of dielectric powders described in [39] (% mass): SiO_2_ (45.91%) + Al_2_O_3_ (23.68%) + TiO_2_ (0.58%) + FeO (8.06%) + MgO (6.05%) + CaO (15.71%). In these experiments, magnesium oxide was replaced by pure magnesium (completely, 50% or 10%), as was described in patent [43] for the generation of ensembles of levitating particles. The particle size distribution was consistent with the granulometric spindle of lunar regolith. The particles of the initial mixture were of irregular shape with sharp edges [42,43]. (Microphotographs and size distributions of the initial laboratory regolith mixture are shown in [42]).

### 2.2. Experimental Technique

Experiments on the action of clouds of charged microparticles obtained from regolith on a stainless steel surface were carried out at the plasmachemical gyrotron stand (gyrotron power up to 400 kW, radiation frequency of 75 GHz) at the plasma physics department of the Prokhorov General Physics Institute of the Russian Academy of Sciences. The scheme of the experiment, the specially designed plasmachemical reactor, and the microwave and video diagnostics, were described in detail in [40,42,43].

A 50 cm-long quartz cylinder with open upper end was inserted in the plasmachemical reactor. All experiments were carried out in air at atmospheric pressure. Figure 1a,b show photographs of the quartz cylinder with the grips for stainless steel plates (view from the side nozzle and top view, respectively). The metal plates were placed at a height of about 3–4 cm above the regolith powder surface (Figure 1c).

As a result of the processing, secondary materials are deposited on the stainless steel plates, the grips, and the walls of the quartz cylinder. To analyze the structure and element composition of the obtained materials, raster electron microscopy and energy dispersive X-ray spectroscopy methods were used. To analyze the structure and element composition of the stainless steel samples and the powders deposited on the walls of the quartz tube (above the placement of the samples), a JEOL JSM-6390LA scanning electron microscope (produced by JEOL Ltd., Tokyo, Japan) with an EX-54175JMX EDS detector (energy-dispersive spectrometer by JEOL Ltd., Tokyo, Japan) was used.

The preliminary DBR treatment was applied to one half of the plates to obtain initial samples with a modified surface, which can be further used for comparative analysis after processing in the gyrotron discharge. Currently, DBR is a known method of surface cleaning (see, e.g., [51,52]). After DBR treatment, the surface of the plates becomes smoother, the number of irregularities and microcracks decreases, traces of metal rolling become visible, and small scratches appear on the surface. The plates consist mainly of steel with the addition of chromium, aluminum, silicon, nickel, etc. The element composition of the plates is given in the first six columns of Tables 1–3. The treatment of the plate surface by low-temperature plasma did not lead to the appearance of new or the disappearance of existing components in its elemental composition but only to small changes in their percentage composition.

The microwave radiation of the gyrotron was supplied to the powders mixture from below through a quartz window with pauses between the microwave pulses no shorter than 20 s. The microwave radiation power was 150–400 kW and the pulse duration was 2–8 ms. The diameter of the microwave beam in the interaction zone with the powders was 8 cm. The microwave radiation energy required to generate the ensembles of levitating particles in these regimes was 1.5–3 kJ.

The development of plasmachemical processes in the reactor both during and after the microwave pulse of the gyrotron was controlled visually by the Fastec Imaging IN250M512 high-speed camera [53] and three optical Ava-Spec spectrometers operating in the range 250–920 nm [40]. Spectrometric measurements were carried out not only from the lower and upper surfaces of the powders but also through the side nozzle ~10 cm above the upper surface of the powders. Each spectrometer recorded 100 spectra at intervals of 4 ms starting from the leading edge of the microwave radiation pulse. The temperature of the powder surface was calculated using the spectral continuum, the plasma temperature between the powder particles and in the layer above the surface was determined from the radiation of atomic and ionic lines (usually, at the initial stage of the processes), and the gas temperature in the reactor was calculated from the vibrational spectra of two-atom molecules.

The observations of the evolution of the discharge glow and particle spread in time were carried out using Activecam AC-D1020 (DSSL Company, Moscow, Russia) and FastecImaging IN250M512 (Fastec Imaging, San Diego, CA, USA) cameras. The second camera recorded frames at intervals of 4 ms and was synchronized with the operation of the two spectrometers whose spectra were used to estimate the temperature of the lower surface of the powder, the gas, and the plasma above the powder.

## 3. Results

### 3.1. Levitating Particles

At microwave pulse energy 1–3 kJ, multiple microwave breakdowns between regolith particles lead to the appearance of plasma inside the powder mixture, which helped the absorption of microwaves and heating of the mixture. These processes continued for 1–2 ms and led to the generation of intermediates (atoms, ions, and molecules of secondary substances) and their escape into the reactor volume above the regolith surface. Above the surface, a thin plasma layer was created that absorbed the microwave radiation. The almost total absorption of microwaves was accompanied by the development of chain plasmachemical reactions both inside the powder and above its surface that led to increased absorption of microwaves and heating of the powder. The rapid escape of electrons led to the charging of the upper surface of the powders and escape of light charged dust from the initial regolith surface into the reactor volume by the Coulomb mechanism [40]. After switching off the gyrotron (the length of the microwave pulse does not exceed 10 ms), the initiated processes continued in the reactor.

In the plasma-gas mixture with escaped particles above the powder surface, interaction of intermediate products with initial reagents takes place, which produces new intermediates. A self-propagating high-temperature synthesis of secondary substances is observed, which is accompanied by heat release and is repeated multiple times by the chain mechanism, generating large quantities of final and intermediate products. The duration of the main stage of synthesis of the chain process reaches fractions of a second and it can go into the self-oscillating regime where its duration exceeds 10 s [54]. Such processes are clearly seen from the bright glow in the reactor, which resembles burning. In Figure 1c, the stainless steel plates are shown over the background of such a glow 8 ms after the switch-off of the gyrotron pulse. Over the background of the glow, relatively large and also glowing particles are seen. The clouds of levitating glowing particles were observed in the reactor above the powder surface for up to one second (the duration of operation of our camera). (At the quenching stage of the chain process, cooling of all components takes place in the reactor, followed by subsequent decrease of emittance (the dynamic range of the camera becomes insufficient). At the same time, the deposition of secondary materials on the quartz cylinder walls and port windows (dusting) makes video recording more difficult.) However, in the first 2–3 frames, they are undistinguishable over the strong glow. Microphotographs and size distributions of laboratory regolith are shown in Figure 2 of work [42]. Figure 2 shows the photographs of levitating particles in the reactor at different times during 64 ms after the gyrotron pulse switch-off (in a discharge without the stainless steel plates). The duration of the synthesis stage of the chain reactions in the reactor was three orders of magnitude longer than the duration of the microwave pulse of the gyrotron, and during the first second after switching off the gyrotron, the suspension of dust particles lifted to tens of centimeters above the powder surface (the height of the reactor tube was 0.5 m, and the particles levitated above its open end).

The energy released in the exothermic reactions after the switch-off of the gyrotron exceeds the energy of the initiating microwave pulse. The medium in which the plasmachemical reactions occur is at non-equilibrium. The temperature of the surface of the powder exceeds 4000 K (not only melting but also evaporation of high-melt metals occur at this temperature), the temperature of the nonequilibrium plasma fraction near the powder surface reaches 0.5–0.7 eV (6000–8000 K), and the gas temperature in the reactor exceeds 3530 K (at which aluminum oxide AlO is produced), and it can also exceed 5000 K (by the vibrational spectrum of TiO) at different stages of burning. The chain reactions develop in the central part of the reactor far from the walls, where the maximum of the field is located (microwaves focusing in the beam waist). For the newly created substances, the initial charged dust particles serve as crystallization centers on which substances are deposited and melted at the same time. (Microphotographs and size distributions of laboratory regolith after microwave discharge are shown in [42]).

At all stages (initiation, synthesis, and decay) of the chain plasmachemical reactions, regolith particles with spheroid shape, melted smooth surface and diameter from 0.1 to 1000 µm (to the difference from the initial shape from the initial regolith with sharp edges) are deposited on the quartz tube walls along its entire height. The chemical deposition and size distribution of the particles of deposited mixture repeat the initial powder mixture of regolith, the difference being the melted shape of the deposited particles (a detailed analysis of physical-chemical properties of secondary particles was made in [42]).

### 3.2. Modification of Stainless Steel Samples in Simulation Experiments

Samples of 309L stainless steel were placed in the reactor for simulation experiments on the effect of levitating regolith particles and located above the regolith powder layer, as is shown in Figure 1c. Both untreated and DBR-treated plates were processed simultaneously in three successive gyrotron pulses with energy that exceeded the threshold of development of all the processes described above (radiation power 300 kW, microwave pulse duration 4 ms, pause between pulses 1 min). Thus, the plates were in the cloud of levitating regolith particles during each gyrotron pulse (thrice total in each cycle of chain reactions accompanied by the spread of particles). Figure 3 shows the pictures of the stainless steel surface and its element composition after such interaction.

The surfaces of the samples and their element composition change after the simulation interaction with levitating particles in the reactor. The deposited layer on the sample without initial treatment was denser, and deposition of spheroids of different size was seen on it. Due to the large quantity of deposited substances, substantial change in the element composition of the sample was observed. On the sample that was initially treated by the DBR, the deposited layer is thinner, but the element composition also changed. On both samples, silicon was deposited, but its amount was substantially higher on the dirty untreated surfaces than on the DBR-cleaned ones.

Let us consider in more detail the change of the surfaces of treated and untreated samples.

#### 3.2.1. Modification of the Surface of Stainless Steel without Prior Treatment 

Table 1 shows the element composition of the surface of the untreated stainless steel sample (analysis over the entire photograph in Figure 3) before and after the simulation experiments.

In the sample before interaction, more than half of the surface composition was taken by iron, the quantity of silicon was low, and oxygen and magnesium were absent. After processing in the simulation discharge with regolith, the surface of the sample was covered by particles that mainly consisted of silicon oxides and oxides of different metals. During analysis of the entire surface of the deposit, no aluminum was found (within the error margin of the analysis). The elemental composition indicated that particles containing silicon and alkali-earth metals were deposited on the surface of the plates. The dense coating of the surface of the sample with melted spherical regolith particles is confirmed by the photograph of the plate with strong magnification that is shown in Figure 4a.

Estimates of the linear size of the particles and their size distribution were made using electron photographs of the plates by the ImageJ software package [55] (the particles were defined as ellipsoids). The particle size distribution is shown in Figure 4b. Note that the number of small-sized particles with diameters below 1 µs is overestimated because the software counts the irregularities of large particles as separate particles. A substantial number of particles with linear size over 10 µm was found.

Separate particles reach the size of over 20 µm. Table 2 shows the element composition of a 1-µm-diameter spot on such a large particle. The large spherical particle consists of silicon oxide with small quantities of other oxides (including aluminum oxide). In such particles, carbon is absent. The magnified image of the plate shows a complicated structure of the deposited particles with sizes of several microns (the inset in Figure 4a). 

#### 3.2.2. Modification of the Surface of Stainless Still with Preliminary Treatment of the Sample Surface by the DBR 

Table 3 shows the element composition of the surface of the stainless steel sample that was initially treated by the DBR before and after the simulation experiment (analysis was carried out over the entire frame shown in Figure 3), i.e., the plates were in the levitating clouds of regolith particles during three gyrotron pulses.

In the sample before interaction, over half the composition of the surface of stainless steel initially treated by the DBR consists of iron and among the admixtures, the quantity of silicon is low and oxygen and magnesium are absent. After processing in the simulation discharge with regolith, the surface of the metal plates is not fully covered by deposited particles. This surface consists of iron weakly covered by deposited materials. Element analysis shows the appearance of magnesium and also metal oxides and silicon. By contrast with the plates that were not treated by the DBR and became covered by spheroids of different sizes, the DBR-treated plates were covered by particles of only two types.

Figure 4c shows these two types of particles: amorphous and spherical (inset in the upper left corner). Amorphous particles (in the center) consisted of conglomerates of particles of irregular shape, but they could also include spherical particles. The elemental composition of the particles was close to that of regolith powder and not of stainless steel. In the particle size distribution, the relative deposition of small particles decreased (this is also an estimate from above that includes the irregularities of large particles) compared to the size distribution of particles deposited on untreated samples (Figure 4b,d).

The inset in Figure 4c shows the photograph of a spherical particle with diameter of about 15 µm with irregular surface. Table 4 shows its element composition, which is dominated by the aluminum, silicon, and magnesium oxides, i.e., powders that compose the regolith, and shows notable absence of iron.

## 4. Discussion

During the action of powerful microwave radiation on metal–dielectric powder mixtures whose composition and size distribution are the same as those of lunar regolith samples, levitating ensembles of charged particles are created. The laboratory simulation experiments are based on the similarity between the physical (microwave breakdown, Coulomb scatter of charged particles) and chemical (exothermic self-sustained reactions) processes that can develop in regolith when it attains sufficient energy, e.g., during bombardment of the Moon’s surface with micrometeorites and when it is irradiated by the microwave radiation of a powerful gyrotron. The use of a new method is shown, which allows one to imitate under Earth’s conditions the physical-chemical conditions on the Moon that occur upon the impact of meteorites with the layer of lunar regolith.

Simulation experiments are presented in which levitating particles of regolith are deposited on stainless steel. It is shown that the structure of the layer of particles deposited on the surface of the samples changes if the surface is initially treated by low-temperature plasma. After simulation experiments, the untreated stainless steel surface is densely covered by the regolith particles in the shape of spheroids of different sizes and shapes, the majority of which is particles with the linear size of several microns. At the same time, the surface of samples that were initially treated by the DBR is not fully covered by regolith particles, which instead form a rare coating that consists of large particles and agglomerations of particles with sizes from several microns to 10 µm whose composition is the same as that of regolith. Thus, it was shown on the example of stainless steel samples that the simulation experiments are sensitive to the quality of the surface, which allows one to choose the materials for space technology that is planned to be used during Moon missions that will be only weakly affected by lunar dust.

In further experiments, we consider the possibility of using the newly developed method in simulation experiments on the deposition of regolith particles on samples from other materials, both metals and dielectrics.

## 5. Conclusions

In this paper, a new microwave laboratory method is presented, which allows one to imitate under Earth’s conditions the physical-chemical parameters of the dusty plasma on the Moon that is created upon the impact of meteorites with the layer of lunar regolith. This method makes it possible to test materials for space technology for future Moon missions.

## Figures and Tables

**Figure 1 materials-14-06472-f001:**
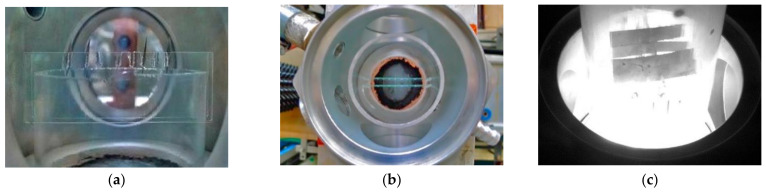
Photographs of the plasmachemical reactor with inserted quartz cylinder and grips for stainless steel plates: (**a**) view from the side nozzle, (**b**) top view, and (**c**) placement of stainless steel plates in the grips. Photograph (**c**) was taken 8 ms after the switch-off of the gyrotron pulse against the background of the processes in the reactor.

**Figure 2 materials-14-06472-f002:**
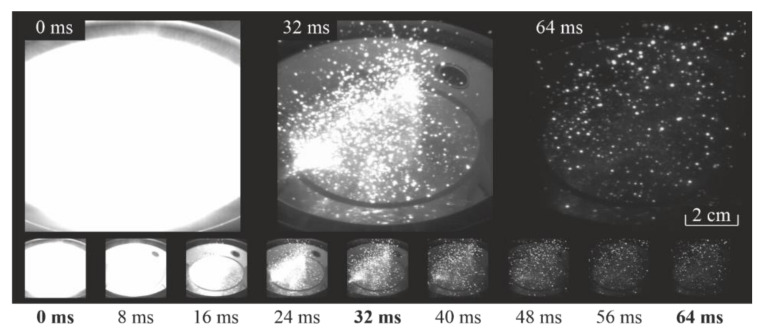
Levitating particles at a distance about 30 cm above regolith surface after initiation of plasmachemical reactions (time t = 0 corresponds to the switch-off of the gyrotron).

**Figure 3 materials-14-06472-f003:**
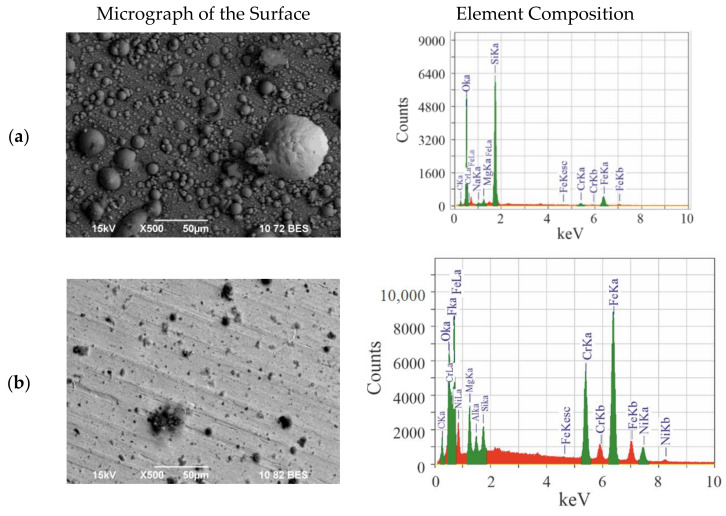
Change of the surface of stainless steel after three cycles of chain processes in simulation experiments. Photographs of the deposited secondary materials (**left**) and their element composition (**right**). The secondary materials are deposited as spheroids of different sizes. (**a**) Sample of untreated surface of stainless steel with regolith deposits and (**b**) sample of DBR-treated surface of stainless steel with regolith deposits. The element composition of the untreated surface reflects mainly the deposited particles, while the composition of the DBR-treated surface includes the components of stainless steel (CrKα, FeKα lines).

**Figure 4 materials-14-06472-f004:**
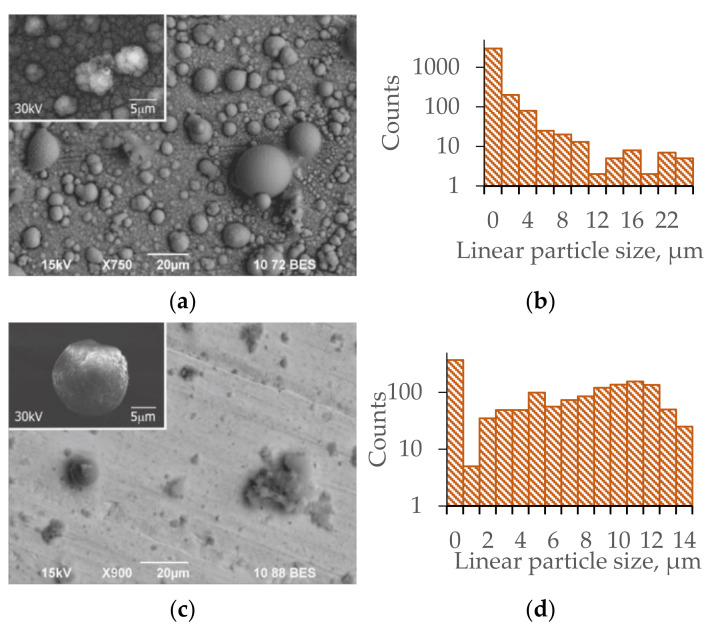
(**a**) Photograph of the plate without initial treatment with deposited particles with resolution of 20 µm. The inset shows a fragment with separate particles at resolution of 5 µm; (**b**) distribution of particles shown in plate (**a**) over linear sizes; (**c**) photograph of the plate that was initially treated by the DBR with deposited particles with resolution of 20 µm. The inset shows a fragment with a single particle at resolution of 5 µm; and (**d**) distribution of particles shown in plate (**c**) over linear sizes.

**Table 1 materials-14-06472-t001:** Elemental composition of the surface of untreated stainless steel sample before and after simulation interaction.

Element Composition of the Surface of the Untreated Stainless Steel Sample before Simulation Interaction						Element Composition of the Surface of the Sample after the Interaction			
Element, Series	Electron Beam Energy, keV	Counts	Mass %	Discrepancy %	Atom %	Counts	Mass %	Discrepancy %	Atom %
C, K	0.277	12,606.46	2.51	0.02	10.06	1306.13	2.85	0.04	5.39
O, K	0.525	-	-	-	-	44,091.99	37.22	0	52.77
Na, K	1.041	-	-	-	-	599.31	0.39	0.50	0.38
Mg, K	1.253	-	-	-	-	1732.13	1.10	0.19	1.02
F, K	0.677	41,627.76	2.90	0.04	7.36	-	-	-	-
Al, K	1.486	1757.53	0.11	1.41	0.19	-			
Si, K	1.739	3679.56	0.23	0.73	0.39	60,633.50	41.47	0.01	33.48
Cr, K	5.411	146,213.40	25.84	0.02	23.94	1101.81	2.14	0.42	0.93
Fe, K	6.398	249,163.70	58.86	0.01	50.78	5716.93	14.83	0.08	6.02
Ni, K	7.471	23,340.19	7.70	0.14	6.32	-	-	-	-
Nb, L	2.166	7069.07	1.86	0.41	0.96	-	-	-	-
Total			100		100		100		100

**Table 2 materials-14-06472-t002:** Element composition of the 1 µm-diameter spot on the large spherical particle (inset in Figure 4a).

Element, Series	Electron Beam Energy, keV	Counts	Mass %	Discrepancy %	Atom %
O, K	0.525	6615.14	41.54	0	57.71
Mg, K	1.253	496.36	2.34	0.11	2.14
Al, K	1.486	1220.90	6.06	0.05	4.99
Si, K	1.739	7350.18	37.39	0.01	29.59
Ca, K	3.690	389.78	3.35	0.20	1.86
Fe, K	6.398	483.90	9.34	0.15	3.72
Total			100		100

**Table 3 materials-14-06472-t003:** Elemental composition of the surface of stainless steel sample treated by the DBR before and after simulation interaction.

Element Composition of the Surface of the Stainless Steel Sample Treated by the DBR before Simulation Interaction						Element Composition of the Surface of the Sample after the Interaction			
Element, Series	Electron Beam Energy, keV	Counts	Mass %	Discrepancy %	Atom %	Counts	Mass %	Discrepancy %	Atom %
C, K	0.277	6454.62	3.12	0.02	12.12	6966.39	2.74	0.02	9.58
O, K	0.525	-	-	-	-	28,568.73	4.34	0.02	11.40
Mg, K	1.253	-	-	-	-	22,053.37	2.51	0.05	4.34
F, K	0.677	19,365.27	3.28	0.03	8.06	18,792.26	2.59	0.04	5.73
Al, K	1.486	1599.29	0.24	0.65	0.41	8201.06	0.99	0.14	1.53
Si, K	1.739	1559.22	0.24	0.72	0.39	12,543.42	1.54	0.10	2.31
Cr, K	5.411	57,984.29	24.90	0.02	22.35	64,611.65	22.58	0.02	18.23
Fe, K	6.398	104,989.70	60.28	0.01	50.36	118,733.20	55.46	0.01	41.70
Ni, K	7.471	9911.21	7.95	0.14	6.31	11,103.42	7.24	0.13	5.18
Total			100		100		100		100

**Table 4 materials-14-06472-t004:** Elemental composition of the 1 µm-diameter spot on the large spherical particle (inset in Figure 4c).

Element, Series	Electron Beam Energy, keV	Counts	Mass %	Discrepancy %	Atom %
C K	0.277	759.22	1.89	0.09	3.24
O K	0.525	41,280.86	39.75	0.01	51.16
Mg K	1.253	18,238.85	13.17	0.03	11.16
Al K	1.486	57,839.12	44.01	0.01	33.58
Si K	1.739	1498.56	1.17	0.4	0.86
Total			100		100

## Data Availability

The data presented in this study is available on request from the corresponding author.

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
