# Peer review of "Microwave Simulation Experiments on Regolith (Lunar Dust) Deposition on Stainless Steel"

_materials, 2021, doi:10.3390/ma14216472_

Round 1

Reviewer 1 Report

The paper discussed the effect of  levitating particles of lunar regolith on the surface of 309L stainless steel plates. The relationships between prior treatment and chemical composition of the surface of workpiece were studied experimentally. It is a good study on this subject.

The structure is complete, and the logic is clear.  The paper's well written and the reviewer has no particular comment.

Please consider the following items in order to be published in this Journal.

page 2: "106 e" ?

Introduction section: In this section, the authors don’t indicate the novelty of their work. What is the innovation of your work when compared with the other researchers?

Figure 3: It is suggested to mark characteristic features in the micrographs.

Can the proposed method be used for the tribological modification of the surface?

What is the practical application of the proposed method for material surface modification?

List of references is dominated by authors from Russian Federation. Have researchers from other countries carried out similar studies?

Author Response

Please find the answers in the attached file.

Kind regards,

Prof. Nina N. Skvortsova

Author Response

(The authors gave the same response as above.)

Reviewer 3 Report

The manuscript "Microwave Simulation Experiments on Regolith (Lunar Dust)
Deposition on Stainless Steel" has been reviewed. It deals with simulation experiments on regolith deposition on stainless steel.

The manuscript is interesting, clear and well organized. English is fine.

I suggest the following minor revisions:

Figs. 4 a), b) and d) are not called in the main text. Pleae check.

The word "error" is not suitable in many tables. Please indicate uncertainty.

I just suggest to split conceptually § 4 Discussion in § 4 Discussion and § 5 Conclusions

Author Response

(The authors gave the same response as above.)
